

# Electrophysiological correlates of aesthetic processing of webpages: a comparison of experts and laypersons

Jens Bölte[1], Thomas M. Hösker[1], Gerrit Hirschfeld[2] and Meinald T. Thielsch[1]

[1] Department of Psychology, Westfälische Wilhelms-Universität Münster, Münster, Germany
[2] Faculty of Business Management and Social Sciences, University of Applied Sciences Osnabrück, Osnabrück, Germany

## ABSTRACT

We investigated whether design experts or laypersons evaluate webpages differently. Twenty participants, 10 experts and 10 laypersons, judged the aesthetic value of a webpage in an EEG-experiment. Screenshots of 150 webpages, judged as aesthetic or as unaesthetic by another 136 participants, served as stimulus material. Behaviorally, experts and laypersons evaluated unaesthetic webpages similarly, but they differed in their evaluation of aesthetic ones: experts evaluated aesthetic webpages as unaesthetic more often than laypersons did. The ERP-data show main effects of level of expertise and of aesthetic value only. There was no interaction of expertise and aesthetics. In a time-window of 110–130 ms after stimulus onset, aesthetic webpages elicited a more negative EEG-amplitude than unaesthetic webpages. In the same time window, experts had more negative EEG-amplitudes than laypersons. This patterning of results continued until a time window of 600–800 ms in which group and aesthetic differences diminished. An interaction of perceiver characteristics and object properties that several interactionist theories postulate was absent in the EEG-data. Experts seem to process the stimuli in a more thorough manner than laypersons. The early activation differences between aesthetic and unaesthetic webpages is in contrast with some theories of aesthetic processing and has not been reported before.

## INTRODUCTION

Humans appreciate aesthetic entities in various contexts: entities might be created for aesthetic and intellectual purposes only as in fine arts. Aesthetic properties of a product might serve to attract customers. Humans evaluate the aesthetic properties of everyday objects such as tableware, TV-sets, or food. Aesthetic judgments play a role in many aspects of human life (*Hoyer & Stokburger-Sauer, 2012*) differing between cultures, situations, educational background, expertise and other individual properties (*Jacobsen, 2006*).

We focus in our research on the differences that experts and laypersons might show in evaluating the aesthetic quality of webpages. Experts and laypersons differ in their aesthetic judgments in various domains such as art, music, movies, software code, or even facades

Corresponding author
Jens Bölte, boelte@uni-muenster.de

of houses (*Hasse & Weber, 2012*; *Kozbelt et al., 2012*; *Müller et al., 2009*; *Silvia, 2013*; *Silvia & Berg, 2011*). Websites are a modern medium of communication serving nearly every aspect of human living. They are evaluated not only in terms of functionality but also in terms of their aesthetic properties (*Lavie & Tractinsky, 2004*; *Moshagen & Thielsch, 2010*). While electrophysiological responses to music, for instance, have been described (*Müller et al., 2009*; *Müller et al., 2010*), it is unclear whether similar event-related potentials will be present when evaluating webpages.

## Aesthetics of websites

Aesthetics has become a core construct in research on human-computer interaction. Similar to other everyday stimuli such as logos (*Handy et al., 2010*), webpages (*Lindgaard et al., 2006*; *Thielsch & Hirschfeld, 2012*; *Tractinsky et al., 2006*) are spontaneously processed at an aesthetic level (for an overview see *Tuch et al., 2012a*). Yet website aesthetics, i.e., the "immediate pleasurable subjective experience that is directed toward an object and not mediated by intervening reasoning" (*Moshagen & Thielsch, 2010*, p. 690), is not only influencing early perceptive stages but also later processing, such as the formation of behavioral intentions (*Moshagen & Thielsch, 2010*; *Thielsch, Blotenberg & Jaron, 2014*). Aesthetics has an impact on various constructs: for example, perceived usability, credibility, satisfaction, preference, urge to buy impulsively, or intention to revisit (for overviews see *Lee & Koubek, 2012*; *Moshagen & Thielsch, 2010*; *Tuch et al., 2012b*).

The current research suggests that aesthetic responses to webpages occur immediately at first sight (e.g., *Lindgaard et al., 2006*; *Lindgaard et al., 2011*; *Thielsch & Hirschfeld, 2012*; *Tractinsky et al., 2006*). This is not only in line with results of prior research on the aesthetics of art, but as well of high practical relevance, as users' first impressions are very relevant for the decision to explore a particular website more thoroughly or leave it for another (see *Thielsch, Blotenberg & Jaron, 2014*). However in contrast to pieces of art, the major purpose of a website is not to elicit an aesthetic experience. Websites are mostly designed to provide some kind of information, typically in an interactive manner (see *ISO, 2006*; *Thielsch, Blotenberg & Jaron, 2014*).

## Models of aesthetic processing

*Fechner (1876)* pioneered the empirical investigation of aesthetic processing and started the empirical search for object properties that determine the subjective aesthetic evaluation. Following this approach, a number of properties have been discussed, e.g., balance and proportion (*Arnheim, 1974*; *Birkhoff, 1933*; *Fechner, 1876*), novelty and prototypicality (*Hekkert, Snelders & Wieringen, 2003*; *Hekkert & Van Wieringen, 1990*), contrast and clarity (*Gombrich, 1995*; *Solso, 2003*) as well as the relation between object features, particularly between simplicity and complexity (*Birkhoff, 1933*; *Von Ehrenfels, 1890*; *Eysenck, 1941*). In recent years, interactionist perspectives on aesthetics have been put forward focusing on the interplay between the observer and the object (e.g., *Berlyne, 1971*; *Chatterjee, 2004*; *Jacobsen, 2006*; *Jacobsen, 2010*; *Leder et al., 2004*; *Leder & Nadal, 2014*; *Reber, Schwarz & Winkielman, 2004*). This perspective promotes the idea that aesthetic processing operates on distinct processing stages.

According to the framework by *Leder et al. (2004)* and *Leder & Nadal (2014)*, aesthetic processing follows five stages: perception, implicit classification, explicit classification, cognitive mastering and evaluation. Affective states accompany cognitive processing in an interactive manner. The cognitive processing results in two outcomes: first, the aesthetic judgment reflecting the degree by which the object in question meets the normative standards of aesthetics. Second, the aesthetic emotion or appraisal is characterized by the affective experiences during the aesthetic processing.

Whereas this framework of aesthetic appreciation and aesthetic judgments (*Leder et al., 2004*) emphasizes psychological processes, the model of visual neuroaesthetics by *Chatterjee (2004)* focuses on the neuronal basics of aesthetic processing. Early and intermediate visual processing steps are similar for aesthetic and non-aesthetic processing in this model. During early vision, basic stimulus parameters such as color, shape and contrast are processed (*Marr, 1982*). Thereafter during intermediate vision, the stimulus parameters are grouped together. At the latest processing stage, object recognition, affective and aesthetic evaluation take place. Aesthetically relevant object features trigger attention processes during early and intermediate stages of visual processing. Thereby, the processing of the aesthetic characteristics is enhanced in later phases of visual processing in terms of a feed forward system (*Chatterjee, 2004*).

## EEG-studies on aesthetic processing

EEG-studies are a valuable tool to shed some light on the timing of aesthetic processing. The comparison of aesthetic and descriptive assessments of geometric patterns showed that aesthetic processing is a two-step process characterized by an early frontal negative potential (ERAN) between 300 and 400 ms after stimulus onset that is more negative for unaesthetic stimuli compared to aesthetic ones. This difference between stimuli is diminished when participants have to judge whether the stimulus is symmetric or not. The second step is associated with a late positive potential (LPP) in central and parietal electrodes between 440 and 880 ms peaking around 600 ms after stimulus onset. Again, the LPP is more positive going for aesthetic images than for unaesthetic images and only found when participants have to rate the aesthetics of an image. The ERAN supposedly reflects impression formation, while the LPP indicates evaluative stimulus categorization (*Höfel & Jacobsen, 2007a*; *Höfel & Jacobsen, 2007b*; *Jacobsen & Höfel, 2003*).

*De Tommaso et al. (2008)* first asked their participants to categorize targets (images of paintings or geometric figures) as beautiful, neutral or ugly and to ignore standard stimuli (a green screen). In a second session, they asked participants to categorize targets as known or unknown. ERPs in these two sessions were scrutinized for differences between beautiful, neutral and ugly pictures. In the first session, the authors found a main-effect for the rated aesthetics of the pictures on the N2b amplitude—with neutral, beautiful and ugly pictures showing a larger N2b compared to the standard stimuli. No interactions with task or electrode were found. For the P3b, all main effects were significant indicating more positive responses to beautiful pictures, the artistic task and at parietal electrodes. In Experiment 2, N2b-amplitudes showed no modulation due to aesthetic value. However, beautiful target stimuli elicited a larger P3b than ugly or neutral target stimuli, all of which

elicit a larger P3b than standard stimuli. The authors argue that task differences bring about the different results; however, they did not formally test for differences between the tasks. Together these studies showed that both early (N2b) as well as late effects occur (P3b, ERAN, LPP), at least when the task is to evaluate the aesthetic value of the stimuli.

Using an oddball-paradigm, *Wang et al. (2012)* investigated the processing of aesthetic and unaesthetic images without explicitly instructing their participants to form aesthetic judgements. They observed a P2 component, the only component that they report, that was most pronounced at frontal sites and enhanced for less beautiful pendants in comparison to more beautiful pendants. A P2 supposedly indicates early affective evaluation, for instance of words or pictures that elicit negative feelings (*Huang & Luo, 2006*). Therefore, Wang et al. concluded that the P2 they observed reflects early automatic emotional processes that accompany aesthetic processing. Alternatively, *Carretié et al. (2001)* suggested that attentional processes might be involved in the P2. Given that Wang et al. used an oddball-paradigm that is suited to investigate automatic attentional processes, this alternative interpretation cannot be ruled out. *Carretié et al. (2004)* suggest that the P2 does not only reflect low-level feature processing, but rather shows the effect of attentional and emotional characteristics of the stimuli.

## Individual differences: experts versus laypersons

The above-mentioned empirical findings reveal the impact of stimulus properties on the different suggested processing steps (*Chatterjee, 2004*; *Jacobsen, 2006*; *Leder et al., 2004*). However, varying subject characteristics such as age (*Thielsch, 2008*), gender (*Cela-Conde et al., 2004*; *Tuch, Bargas-Avila & Opwis, 2010*) or domain specific expertise that could influence aesthetic processing were not taken into account (*Chevalier & Ivory, 2003*; *Park, Choi & Kim, 2004*). Especially, the model by Leder et al. that refers to implicit or explicit knowledge in long-term memory allows predicting differences in processing by experts and laypersons.

In the context of face perception research, it has been shown that expertise for face-like objects can be acquired by training (e.g., *Gauthier & Tarr, 1997*) and that the processing of faces or other well-trained materials goes along with distinct brain activity (*Bentin et al., 1996*; *Gauthier et al., 2000*; *Tanaka & Curran, 2001*). Thus, expertise for specific objects can be acquired and is reflected in distinguishable cortical activity (*Scott et al., 2006*).

For example, architects show higher hippocampus, precuneus, orbitofrontal cortex and gyrus cinguli activity than laypersons when asked to rate the aesthetics of buildings (*Kirk et al., 2009*). Moreover, designer and laypersons differ in their cortical activity during a design task (*Kowatari et al., 2009*). Similarly, art specific expertise is accompanied by functional and structural modifications, i.e., higher cortical activation during color processing and higher density of grey matter in area V4 (*Long et al., 2011*). Furthermore, laypersons and experts differ in perceptual exploration (*Hekkert & Van Wieringen, 1996a*; *Hekkert & Van Wieringen, 1996b*), processing of complexity (*Reber, Schwarz & Winkielman, 2004*) and aesthetic preferences (*Hekkert & Van Wieringen, 1996a*; *Hekkert & Van Wieringen, 1996b*). Expertise is presumably characterized by stronger than usual neuronal connections

of specific representations or processes which result in high accessibility of these representations (*Cheung & Bar, 2012*; *Harel et al., 2010*).

*Müller et al. (2010)* investigated the differential processing of short piano sequences by laypersons and experts using EEG. They observed a larger frontal P2 amplitude for experts than for laypersons—possibly the same component as studied by Wang and colleagues *(2012)*—and a larger ERAN, which peaked 200 ms after stimulus onset. Müller et al. assumed that the enhanced P2 reflects extended neural representations for musical stimuli in experts. The larger ERAN indicates a more thorough processing of the stimuli by the experts than by the laypersons because it was only observed with mild harmonic violations.

Using paintings, filtered copies of these paintings and plain-color stimuli as visual stimuli, *Pang et al. (2013)* could show that paintings elicit larger P3b components than their filtered copies or plain-color stimuli in a free-viewing task. Experts, determined by a questionnaire, had smaller P3b/LPC-like bilateral posterior ERPs than laypersons. Pang et al. argued that this reduced activity is a consequence of neural efficiency due to increased practice that is also reflected in non-directive tasks. Clearly, this observation contradicts the one reported by *Müller et al. (2010)*. Müller et al. observed increased neuronal activation for experts in comparison to laypersons. However, various differences between the experiments, visual vs. auditory stimulation, passive viewing vs. explicit judgment, and choice of reference electrodes, prohibit firm conclusions. The idea that expertise is associated with stronger neuronal connections between cortical areas involved in stimulus processing is compatible with both increased as well as decreased activations for experts compared to laypersons (*Cheung & Bar, 2012*; *Harel et al., 2010*). Taken together, the research has revealed several differences between experts and laypersons when it comes to the processing of aesthetic stimuli. It is unknown whether these differences map to the processing stages surmised by current models of aesthetic perception (*Chatterjee, 2004*; *Jacobsen, 2006*; *Leder et al., 2004*).

### Research question

The research presented above shows that experts and laypersons differ in the processing of stimuli, including the evaluation of aesthetic stimulus qualities. Most often music or stimuli constructed for the particular research question were used as stimulus material. We want to extend this research by using commonly experienced stimuli such as webpages. Since studies depending on design, stimuli and task-type resulted in vastly different ERP-components that were modulated by aesthetics and expertise, we tested for differences in those time-windows that corresponded best to the ERP found in this study.

## MATERIALS AND METHODS

### Participants

There were 20 participants, 10 were experts (mean age: 32.2 years, $SD = 12.5$ years, $Min = 22$, $Max = 62$, 5 females) and 10 laypersons (mean age: 31 years, $SD = 11.4$ years, $Min = 20$, $Max = 62$, 5 females). In addition, laypersons were also matched to experts in terms of education. Experts were either professionals in or students of design, graphic design or digital media design. They had a mean experience in the area of 10.8 years ($SD = 10.7$ years). Experts attached more value to the visual-aesthetic product design than

laypersons in the Centrality of Visual Product Aesthetics questionnaire (CVPA; *Bloch, Brunel & Arnold, 2003*; German version *Thielsch, 2008*; experts: $M_{CVPA} = 4.18$, $SD = .41$; laypersons: $M_{CVPA} = 2.9$, $SD = .52$; $t(18) = 6.08$, $p < .001$, $d = 1.6$).

Participants received 10.00 € for their participation. They had unimpaired or corrected-to-normal visual acuity as well as normal color perception. Our study did not require the approval of our local Ethics Committee at the Department of Psychology at the University of Münster as we performed a non-clinical website evaluation study and used only non-invasive measures (ratings, reaction times, EEG). The participants' task was to assess the aesthetic values of webpages. No treatments or false feedbacks were given; no potential harmful evaluation methods were used. Participation was voluntary and participants could drop out at any time without any negative consequences. Informed consent was obtained from all participants. All data were stored using an anonymous ID for each participant.

## Materials

We used 150 webpage-screenshots in the experiment. These screenshots were selected from a larger set of 300 webpage-screenshots that had been rated in terms of their aesthetic attractiveness by another 136 participants (mean age: 25.2 years, $SD = 6.77$ years, 110 females) using the short version of the Visual Aesthetics of Websites Inventory (VisAWI-S, *Moshagen & Thielsch, 2013*). Websites known by more than 25% of the participants were excluded from further analysis. Based on the results, two sets of 75 webpage-screenshots were created differing significantly in aesthetic assessment ($M_{aesthetic} = 5.11$, $SD = .39$; $M_{unaesthetic} = 3.27$, $SD = .45$; $d = 5.76$; $\chi^2 = 146.03$, $p < .001$). The webpage-screenshots come from ten different content domains (download & software, e-commerce, e-learning, entertainment, e-recruiting, information, corporate websites, social software, search engines and web portals). Information about the browser was removed using Adobe Photoshop, Version CS5.1 (Adobe Systems, San Jose, CA, USA). Webpages were $1,280 \times 780$ to $1,264 \times 765$ pixels large (unaesthetic mean: 1,012,742 pixel, $SD$: 5,820, aesthetic mean: 1,013,931 pixel, $SD$: 5,278; Welch $t(146.61) = -1.31$, $p = .192$), resulting in a visual angle of maximally 18.23°. The average luminance per pixel relative to white (white having a value of 1.0, black having a value of 0) did not differ from each other (unaesthetic mean: .785, $SD$: 0.149, aesthetic: .739, $SD$: 0.182; Welch $t(142.63) = 1.677$, $p = .095$). Also, unaesthetic and aesthetic webpages did not differ in contrast (unaesthetic: .261, SD: 0.069, aesthetic: .252, SD: 0.068; Welch $t(147.93$, $p = .408$). Aesthetic and unaesthetic webpages differed in complexity measured in byte (see *Miniukovich & De Angeli, 2015*; *Tuch et al., 2009*; unaesthetic mean: 681 kb, $SD$: 199; aesthetic: mean: 578 kb, $SD$: 147; Welch $t(136.55) = 3.59$, $p < .001$). The later difference might reflect design differences between unaesthetic (e.g., cluttered layout) and aesthetic webpages (e.g., clear, structured layout).[1]

[1]We are aware that visual complexity measured in bytes is a crude measure of complexity. It is a numerical measure ignoring functional and practical aspects as well as the integration by the observer (see *Xing & Manning, 2005*, for a review of definitions of complexity).

## Apparatus

The experiment was controlled by Presentation Version 16.04.25.12 (Neurobehavioral Systems, Albany, CA, USA). Stimuli were presented on a Samsung Sync-Master 2233, $1,680 \times 1,050$ pixels, screen refresh rate 120 Hz. Responses were collected using a Cedrus response-pad RB 830. The EEG was digitized with a sampling frequency of 256 Hz using

32 sintered Ag/AgCL-electrodes placed according to the 10-20 system (Advanced Neuro Technology, Enschede, The Netherlands) and two mastoid electrodes. We used an online low-pass half-power filter of 69.12 Hz and an average reference for recording. Impedance was kept below 5 kOhm. Vertical EOG was measured by placing a bipolar electrode beneath and above the left eye. Horizontal EOG was measured by placing a bipolar electrode at the outer canthus of each eye. AFz was used as ground electrode.

## Procedure

All participants were interviewed via a telephone-interview in which demographic information, handedness, visual acuity, neurological or psychiatric disorders were assessed. Upon arrival in laboratory, participants were informed about the course of the experiment. First, they completed the Centrality of Visual Product Aesthetics questionnaire (CVPA; *Bloch, Brunel & Arnold, 2003*; German version *Thielsch, 2008*). Afterwards, participants were asked to assess the aesthetics of a given webpage.

The experiment started with 10 practice trials. There was a short break following the practice trials for discussing should anything have remained unclear. A fixation cross was presented to 250 ms in screen center, after this a uniformly light-grey screen was displayed for 1,250 ms, followed by displaying the webpage-screenshot for 2,500 ms. Again, a light grey screen was displayed for 500 ms when three exclamation marks appeared to signal the participants to give their aesthetic judgment (aesthetic or unaesthetic) by pressing one of two keys. Participants were instructed to move as little as possible during recording. Specifically, they were asked to refrain from moving their eyes during the presentation of the webpage and blink after the offset of the webpage. There were short breaks of 10–15 s duration every 90–120 s and a longer break of 90–135 s every 7.5–9.0 min. The experiment lasted about 25 min. In total, 150 stimuli (75 aesthetic and 75 unaesthetic) were presented once in random order determined for each participant.

## Data analysis

The EEG-data were re-referenced to linked mastoids. An offline bandpass-filter (half-power: .1–25 Hz) was applied. An EEG larger than $\pm 75$ $\mu$V was considered an artifact. Artifact free epochs of 1,200 ms length with a baseline of 200 ms were defined.

We used only congruent trials to calculate averages. A trial was considered congruent if the participant's aesthetic judgment and the a-priori classification (see above) matched. This resulted in 705 artifact-free aesthetic trials (experts: 292 i.e., 39% of the trials, laypersons: 413 i.e., 55%) and 1136 artifact-free unaesthetic trials (experts: 541 i.e., 72%, laypersons: 595 i.e., 79%) that were averaged. Based on visual inspection of the ERP, we calculated mean voltage per participant and condition in the following time-windows 80 ms–105 ms, 110 ms–130 ms, 150 ms–370 ms, 370 ms–600 ms and 600 ms–800 ms after stimulus onset. Electrodes were grouped into lateral-central position (left: F3, C3, P3, O1; central: Fz, Cz, Pz, Oz; right: F4, C4, P4, O2), and anterior–posterior position (frontal: F3, Fz, F4; central: C3, Cz, C4; parietal: P3, Pz, P4; occipital: O1, Oz, O2). Mean voltages per participants and time-windows were subjected to mixed ANOVAS with the factors Webpage Aesthetics (aesthetic vs. unaesthetic), lateral-central (lateral vs. central), and

**Table 1  Frequency of aesthetic evaluations as a function of conditions.**

| Group | Website | | | |
|---|---|---|---|---|
| | Unaesthetic | | Aesthetic | |
| | Rating | | | |
| | Aesthetic | Unaesthetic | Aesthetic | Unaesthetic |
| Expert | 94 | 656 | 355 | 395 |
| Layperson | 97 | 653 | 458 | 292 |

**Table 2  Summary regression analysis.**

| Effect | ß | SE(ß) | Odds ratio | p-value |
|---|---|---|---|---|
| Intercept | −2.30 | .24 | 0.10 | <.001 |
| Group | −.03 | .23 | .97 | |
| Webpage Aesthetics | 2.56 | .28 | 12.94 | <.001 |
| Group × Webpage Aesthetics | .63 | .22 | 1.88 | <.01 |

anterior–posterior (anterior vs. posterior) as repeated measures factors and Group (expert vs. layperson) as between-group factor.

# RESULTS

## Behavioral data

Experts and laypersons evaluated unaesthetic webpages similarly but differed in their evaluations concerning aesthetic webpages (see Table 1). A generalized linear mixed logistic regression model was calculated (*Barr et al., 2013*; *Jaeger, 2008*).

The factors Group (expert vs. layperson) and Webpage Aesthetics (aesthetic vs. unaesthetic) served as fixed effects. Trial-number, participant and webpage served as random effects. Random slope and random intercepts for participants and webpage were realized. Dependent variable was the aesthetic evaluation by the participant. The stepwise inclusion of the predictor Webpage Aesthetics and the interaction of Group and Webpage Aesthetics resulted in a significant increase of fit ($\chi^2(7) = 97.58$, $p < .001$; $\chi^2(8) = 8.09$, $p = .005$, respectively, see Table 2 for a summary). The Odds ratio for the predictor Webpage Aesthetics changes by a factor of 12.94 as the webpage aesthetics changes from aesthetic to unaesthetic. The significant interaction shows that the Odds Ratio for an aesthetic evaluation changes by a factor of 1.88 depending on group membership but only if an aesthetic webpage is being evaluated. In sum, experts evaluated aesthetic webpages more often as unaesthetic compared to laypersons.

For the sake of completeness and in addition to the error rates and ERP-analyses, we analyzed the reaction times (RTs). Keep in mind that the RTs reflect the endpoint of perceptual, evaluative and decision processes. Furthermore, the RTs were much longer than the analyzed EEG-interval and were not speeded. That precludes a direct comparison of both measurements. RTs and ERPs probably reflect different processes. We used trimmed mean RT (trimming: 10%) per participant, level of expertise (expert, layperson) and

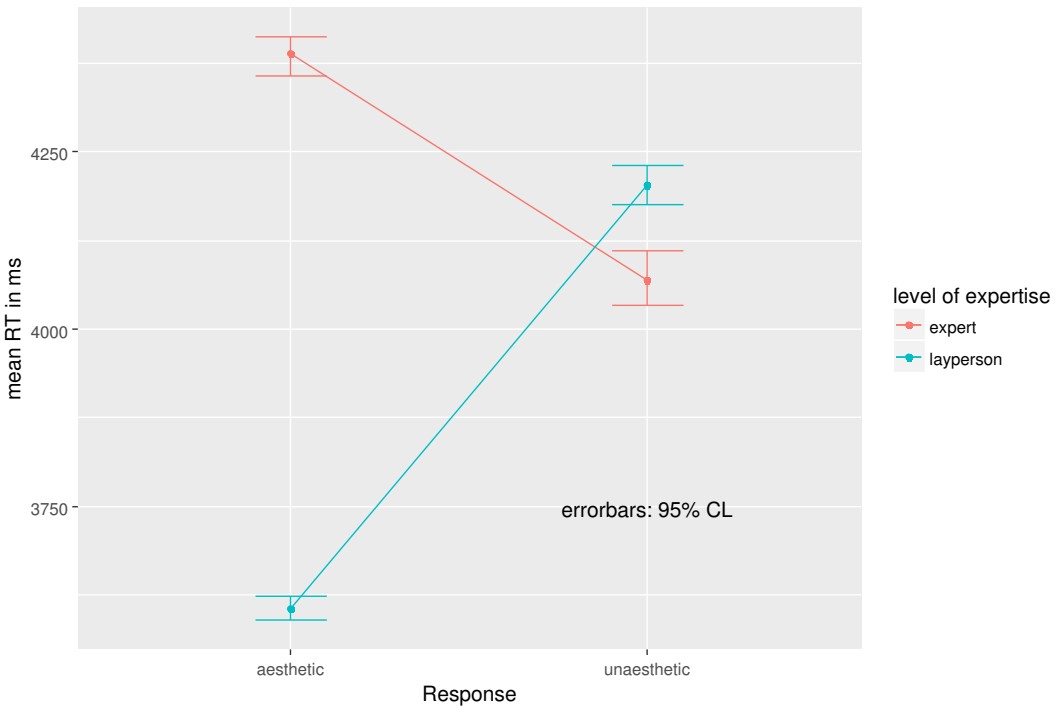

**Figure 1  Trimmed means RT in ms as a function of level of expertise and response.**

**Table 3  Trimmed mean RT in ms and SD (in parentheses) as a function of level of expertise and response.**

| Level of expertise | Response | |
|---|---|---|
| | Aesthetic | Unaesthetic |
| Expert | 4,388 (45) | 4,069 (69) |
| Layperson | 3,605 (31) | 4,202 (45) |

response (aesthetic, unaesthetic) as dependent measure in a mixed ANOVA (within factor: Response; between factor: Group; see Table 3 for descriptive statistics).

The results of the ANOVA showed that experts (mean: 4,229 ms, $SD$: 173) responded slower than laypersons (mean: 3,904 ms, $SD$: 309; $F(1, 18) = 461.38$, $p < .001$, $\eta^2 = .923$. Aesthetic responses (mean: 3,997 ms, $SD$: 403) were faster than unaesthetic ones (mean: 4,136 ms, $SD$: 89; $F(1, 18) = 74.01$, $p < .001$, $\eta^2 = .687$). Both main effects were qualified by a significant interaction ($F(1, 18) = 800.46$, $p < .001$, $\eta^2 = .959$). This interaction reflects the fact that experts and layperson showed different response patterns. While experts were faster in judging a webpage as unaesthetic than as aesthetic, laypersons showed the opposite pattern (see Fig. 1). Laypersons were faster in judging a webpage as aesthetic than as unaesthetic than experts.

## EEG-data

The continuous EEG-signal was split up in five different time windows: 80 ms–105 ms, 110 ms–130 ms, 150 ms–370 ms, 370 ms–600 ms and 600 ms–800 ms after stimulus onset.

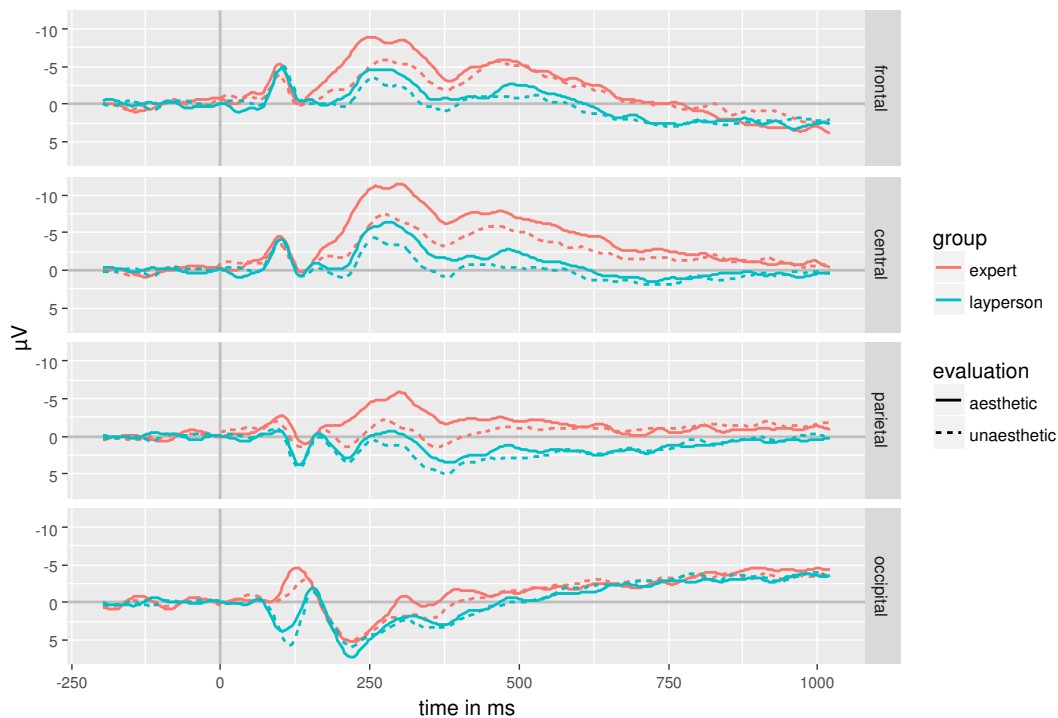

**Figure 2** EEG-signal as a function of expertise, aesthetics and anterior–posterior regions of interests.

**Table 4** Summary of EEG findings in terms of *F*-values.

| Time window | Group (df = 1, 18) | Webpage (df = 1, 18) | Group × Webpage (df = 1, 18) |
|---|---|---|---|
| 80–105 ms (N1) | 4.28 | 1.52 | 1.07 |
| 110–130 ms (P1) | 4.98[*] | 4.49[*] | 0.12 |
| 150–370 ms (N2) | 4.82[*] | 32.43[***] | 2.33 |
| 370–600 ms | 6.22[*] | 6.60[*] | 0.00 |
| 600–800 ms | 0.12 | 0.35 | 0.03 |

**Notes.**
[*]Effect significant at .05 level.
[***]Effect significant at .001 level; df = 1, 18

We used mean amplitude of each time window as dependent variable in separate mixed ANOVAs with the factors Webpage Aesthetics (aesthetic vs. unaesthetic), lateral-central (lateral vs. central), and anterior–posterior (anterior vs. posterior) as repeated measures factors and Group (expert vs. layperson) as between-group factor. Greenhouse-Geisser corrected degrees of freedom are reported in case of a sphericity assumption violation. We only report (near) significant results of experimentally manipulated factors to allow for an easier overview of the results (Table 4). Figures 2 and 3 show the EEG-signal averaged in the regions of interest (Fig. 2: anterior, central, parietal, occipital; Fig. 3: left, central, right).

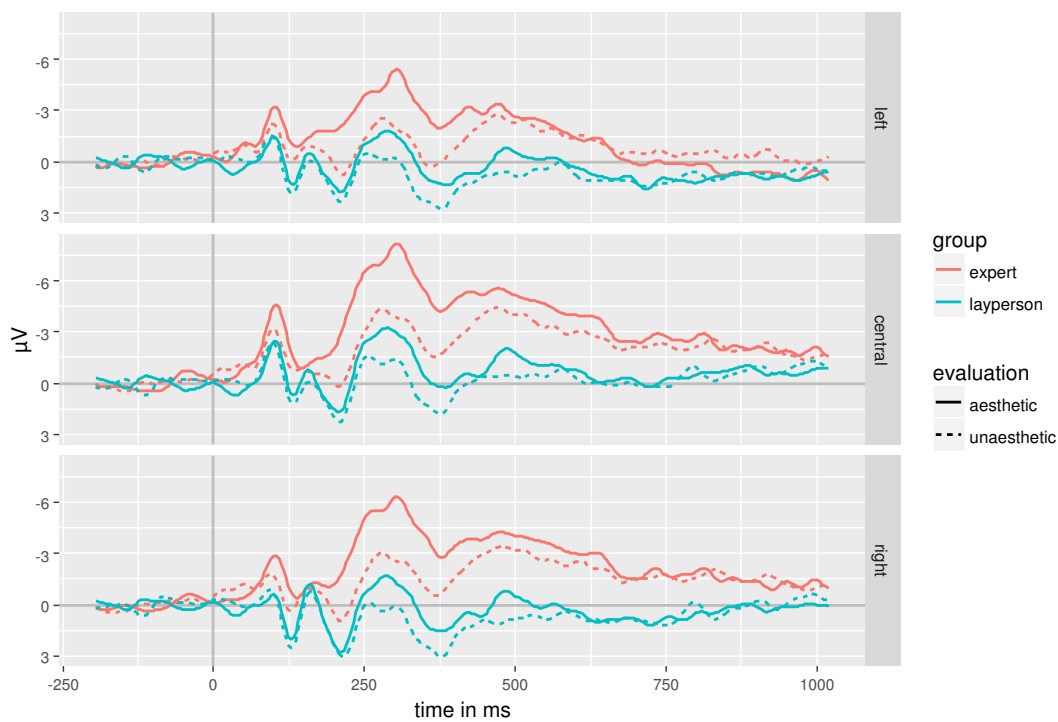

**Figure 3** EEG-signal as a function of expertise, aesthetics and left–right regions of interests.

### *Time window 80 ms–105 ms after stimulus onset*

In this time-window, we see a peak at around 90 ms that is more negative at frontal electrodes than at parietal and occipital electrodes. At occipital electrodes, this peak is negative for experts and positive for laypersons. Nonetheless, the main effect of Group fails significance levels ($F(1, 18) = 4.28$, $p = .053$, $\eta_p^2 = .19$) as does the interaction of Group and Anterior–posterior ($F(1, 18) = 2.64$, $p = .058$, $\eta_p^2 = .13$). Neither the main effect Webpage nor the interaction of Group and Webpage are significant ($F(1, 18) = 1.52$, $p = .233$, $\eta_p^2 = .08$; $F(1, 18) = 1.07$, $p = .314$, $\eta_p^2 = .06$). There are no further significant interactions involving Group or Webpage ($p > .224$ at least).

### *Time window 110 ms–130 ms after stimulus onset*

In this time-window, we see a positive going peak at around 120 ms. It is more pronounced over right than over left electrodes and over parietal and occipital than central and anterior electrodes. Again, there seems to be a differentiation between experts and laypersons at occipital electrodes. Aesthetically evaluated webpages resulted in a more negative going ERP than unaesthetically evaluated webpages ($F(1, 18) = 4.49$, $p = .048$, $\eta_p^2 = .20$). Laypersons have a more positive ERP than experts in this time window ($F(1, 18) = 4.98$, $p = .039$, $\eta_p^2 = .22$). The interaction of Group and Webpage was not significant ($F < 1$). There are no further significant interactions.

### Time window 150 ms–370 ms after stimulus onset

There is a posterior positivity and an anterior negativity in this time window. The main effects of Group and Webpage are both significant while the interaction of Group and Webpage is not significant ($F(1, 18) = 4.82$, $p = .042$, $\eta_p^2 = .21$; $F(1, 18) = 32.43$, $p = .001$, $\eta_p^2 = .64$; $F(1, 18) = 2.33$, $p = .145$, $\eta_p^2 = .11$; respectively). Experts and aesthetic websites evoke more negative ERPs than laypersons and unaesthetic webpages. There are significant interactions of Webpage, lateral-central and anterior–posterior Electrode Positions ($F(2, 36) = 3.31$, $p = .048$, $\eta_p^2 = .15$; $F(1.44, 25.99) = 17.22$, $p < .001$, $\eta_p^2 = .49$). All other interactions are not significant.

Follow up analyses with Webpage and Electrode Position serving as factors show significant effects of Webpage for each lateral-central Electrode Position (left: $F(1, 18) = 29.41$, $p < .001$, $\eta_p^2 = .62$; right: $F(1, 18) = 32.33$, $p < .001$, $\eta_p^2 = .64$; central: $F(1, 18) = 31.04$, $p < .001$, $\eta_p^2 = .63$). Lateral-central Electrode Position and Webpage are also significant (aesthetic: $F(2, 36) = 22.45$, $p < .001$, $\eta_p^2 = .56$; unaesthetic: $F(2, 36) = 11.69$, $p < .001$, $\eta_p^2 = .39$). Concerning the other significant interactions between Webpage and anterior–posterior Electrode Position, follow -up analyses reveal significant effects of Webpage on frontal, central and parietal but not on occipital electrodes (frontal: $F(1, 18) = 39.01$, $p < .001$, $\eta_p^2 = .68$; central: $F(1, 18) = 44.78$, $p < .001$, $\eta_p^2 = .71$; parietal: $F(1, 18) = 26.80$, $p < .001$, $\eta_p^2 = .60$; occipital: $F(1, 18) = .66$, $p = .426$, $\eta_p^2 = .04$). There was also a significant effect of Webpage at anterior–posterior Electrode Position (aesthetic: $F(1.29, 23.20) = 33.72$, $p < .001$, $\eta_p^2 = .65$; unaesthetic: $F(1.38, 24.86) = 25.19$, $p < .001$, $\eta_p^2 = .58$).

### Time window 370 ms–600 ms after stimulus onset

In this time window, the EEG-amplitudes are less pronounced than in the previous time windows and are getting more positive. While at frontal electrodes, the EEG-waves turn from positive voltages to negatives ones, the opposite pattern is seen at occipital electrodes. The differences between the aesthetic and unaesthetic evaluations are reduced compared to previous time windows. With respect to lateral and central electrodes, the EEG becomes less negative at left electrodes, while there is little change in amplitude at central or right electrodes. As before, aesthetic webpages exhibit a more negative EEG as unaesthetic webpages ($F(1, 18) = 6.60$, $p = .019$, $\eta_p^2 = .27$). Experts show a more negative EEG than laypersons ($F(1, 18) = 6.22$, $p = .023$, $\eta_p^2 = .26$). The interaction of Webpage and anterior–posterior Electrode Position is also significant ($F(1.51, 27.1) = 9.98$, $p = .001$, $\eta_p^2 = .36$). The remaining interactions are not significant.

Follow up analyses show that aesthetic webpages differ from unaesthetic webpages at frontal ($F(1, 18) = 6.82$, $p = .018$, $\eta_p^2 = .28$), central ($F(1, 18) = 19.60$, $p < .001$, $\eta_p^2 = .52$) and parietal position ($F(1, 18) = 4.85$, $p = .041$, $\eta_p^2 = .21$), but not at occipital electrode positions ($F < 1$). Moreover, there is a significant effect for anterior–posterior Electrode Position for aesthetic as well as unaesthetic Webpages ($F(1.41, 25.39) = 6.74$, $p = .009$, $\eta_p^2 = .27$; $F(1.64, 29.57) = 5.56$, $p = .013$, $\eta_p^2 = .24$).

### Time window 600 ms–800 ms after stimulus onset

In this time window, the EEG comes from a negative voltage range into a positive one at frontal and central electrodes while the EEG at occipital electrodes shows the opposite

patterns. The EEG at parietal electrodes fluctuates around zero. There are no main effects, but a significant interaction of Webpage and anterior–posterior Electrode Position ($F(1.87, 33.58) = 4.00, p = .030, \eta_p^2 = .18$). Follow up analysis on this interaction reveals a significant effect of anterior–posterior Electrode position for aesthetic as well as unaesthetic Webpages ($F(1.47, 26.44) = 4.61, p = .028, \eta_p^2 = .20; F(1.67, 30.02) = 7.69, p = .003, \eta_p^2 = .30$).

## DISCUSSION

We asked design-experts and laypersons to evaluate the aesthetic properties of static webpages that varied in aesthetic attractiveness. Behavioral responses, i.e., judgements about a webpage's attractiveness, and electrophysiological responses were recorded. The ERPs show early differences between experts and laypersons. We will first summarize behavioral and electrophysiological results before discussing these findings in the context of the models of aesthetic processing presented above.

Participants' behavioral responses indicated that they were more critical than anticipated. Although we presented an even number of aesthetic and unaesthetic webpages, unaesthetic judgements prevailed. Laypersons and experts evaluate unaesthetic webpages similarly. In case of aesthetic webpages, experts seemed to be more critical than laypersons given that they evaluated more webpages as unaesthetic than laypersons did. Nonetheless, the major factor driving the evaluation is the aesthetic quality of the webpage itself, not the person evaluating the webpage as indicated by the 6–7 times larger odds ratio for webpage aesthetic than the odds-ratio for group membership. Given this, expertise is less important than aesthetics in evaluating a webpage.

As the design of this study is not aimed at analyzing RTs, we shortly discuss RTs to ensure completeness but do not want to overemphasize the results. The differences in RTs might be interpreted in a way that experts need more time to make a decision on aesthetics because they scrutinize the aesthetics-relevant details of a webpage more carefully than laypersons do which goes along with underlying attention processes (*Chatterjee, 2004*). Experts are more demanding and have a high standard in regards of the aesthetic quality of a webpage. They have learned to evaluate if a stimulus achieves their standard faster than laypersons. Yet because of their more thorough processing, their decisions are slower. In contrast, laypersons are faster with their decision because they process the webpage in a more superficial way than experts do but they lack a clear standard or expectations, which leads to longer RTs for unaesthetic webpages.

Notice that there was an interaction in the RT-analysis of group and aesthetics while we observed only main effects of group and aesthetics in the ERPs. Thus, it is not likely that the group differences observed in the ERPs are directly linked the RTs. Rather either additional processes not yet reflected in the ERPs contribute to the RTs or the processes reflected by the ERPs undergo further modification. Thus, the processes picked up by ERPs are only the starting point of processing not the end.

Electrophysiological responses showed no significant differences between experts and laypersons in the earliest time-window. However, even at around 100 ms aesthetically evaluated webpages result in stronger ERPs than unaesthetically evaluated ones. In terms

of scalp distribution, this effect is not modulated by electrode position. In contrast, effects starting at around 150 ms and lasting to 600 ms show interactions between electrode position and aesthetic evaluation. An aesthetic evaluation results in a more negative evaluation at frontal, central and parietal electrodes than at occipital electrodes. This pattern reverses independently of aesthetic evaluation in a time window lasting from 600 ms to 800 ms. At the same time, the long-lasting temporal and large spatial distribution of the effects indicate that various, interconnected cortical areas are involved in aesthetic processing (*Kawabata & Zeki, 2004*).

The data are inconclusive with respect to emotion-related potential such as the ERAN and the LPP. The ERAN starts 300 ms to 400 ms after stimulus onset being more negative for unattractive stimuli than for attractive ones (*Höfel & Jacobsen, 2007a*). We see the opposite pattern here. Unaesthetic webpages elicit more positive ERPs starting at a little over 100 ms lasting up to 600 ms. The difference between unaesthetic and aesthetic webpages is more pronounced over frontal to parietal areas than over occipital areas while there is no lateralization of the effect. The direction of the effect and its distribution differ from that of an ERAN. The LPP is often prominent over parietal to central electrodes (*Foti, Hajcak & Dien, 2009*). It is larger for emotional stimuli than for neutral stimuli. We see a decrease in positivity starting at around 300 ms at parietal electrodes. This is accompanied by a decrease in negativity over central electrodes in the same time range. However, a neutral baseline is missing which would allow determining whether these changes are driven by emotional content. In addition, the observed pattern differs from that reported for a LPP (e.g., *Weinberg & Hajcak, 2010*). Thus, it is rather unlikely that we observed a LPP.

## Experts versus laypersons

Modern theories of aesthetic processing assume an interaction of perceiver and object such that experts should process objects differently than laypersons. Our data partly support this assumption. Experts were more critical than laypersons as indicated by the behavioral data. However, a similar interaction of expertise and aesthetic evaluation, for instance greater group differences for aesthetic webpages than for unaesthetic webpages, is absent in the ERPs. This pattern would have been predicted by recent theories of aesthetic processing (*Chatterjee, 2004*; *Leder et al., 2004*; *Reber, Schwarz & Winkielman, 2004*). The main effects of Group and Webpage suggest that the neuronal generators underlying the ERPs are basically the same but experts process the stimuli in a more thorough manner.

Differences between experts and laypersons have been observed in various behavioral studies (e.g., *Hekkert, Peper & Van Wieringen, 1994*; *Hekkert & Van Wieringen, 1996a*; *Hekkert & Van Wieringen, 1996b*; *Pihko et al., 2011*; *Winston & Cupchik, 1992*). For instance, it has been suggested that experts have enhanced associative knowledge that is easily accessible (*Cheung & Bar, 2012*; *Harel et al., 2010*; *Long et al., 2011*; *Tanaka & Taylor, 1991*). Such enhanced memory representations might be activated when processing the presented webpages. Furthermore, motivational and attentional differences might contribute to the observed difference. *Harel et al. (2010)* showed that expertise influences neuronal activation mainly when the expertise is task-relevant. Therefore, it is not a bottom-up, stimulus-driven processing mechanism that differs between experts and

laypersons but rather a top-down (i.e., task demands) modulated processing that results in intensified processing. If this explanation holds, experts and laypersons should not differ when the expertise is not task-relevant as it was in the current study.

### Aesthetics of webpages

Various attempts to obtain an objective measure of the aesthetic value of an object can be found in *Altaboli & Lin (2011)*, *Ngo, Samsudin & Abdullah (2000)* or *Seckler, Opwis & Tuch (2015)*. We used an aesthetics g-factor approach (measured with the VisAWI-S, *Moshagen & Thielsch, 2013*) to divide the webpages in aesthetic and unaesthetic ones. Thus, we are not able to determine which of the suggested dimensions (or which combination of dimensions) brought about the observed behavioral and electrophysiological differences between aesthetic and unaesthetic webpages.

Aesthetic and unaesthetic webpages elicit different EEG-amplitude in an early time-window of 110 ms to 130 ms. ERPs in this time window presumably reflect processing of stimulus properties such as contrast or brightness (e.g., *Luck, 2005*). Often such early visual processes are expressed most over occipital electrodes, which is not what we observe here. We observe a frontal to parietal distribution sparing occipital electrodes. This distributional pattern suggests that not visual properties such as contrast or brightness brought about these differences. However, aesthetic and unaesthetic webpages differed in complexity measured in bytes with unaesthetic webpages being more complex than aesthetic ones. High visual complexity usually results in a more negative evaluation than medium to less complexity (*Tuch et al., 2012a*). Therefore, this early differentiation might reflect visual complexity. But keep in mind, that our complexity measure might be rather crude and probably does not reflect functional complexity. Other properties could bring about the observed difference.

For instance, it might be that (aesthetic) webpages exhibit fractal-like image properties, as do graphic art or natural scenes (*Redies, Hasenstein & Denzler, 2007*). *Redies, Hasenstein & Denzler (2007)* link such fractal-like image properties to the aesthetic perception. A theory of aesthetic processing must take into account human sensory processing. Consequently, web-designer and artists exploit such image-properties because the human visual system has evolved that way. It remains to be determined whether webpages exhibit fractal-like properties and whether they differentiate aesthetic and unaesthetic webpages. Keep in mind that fractal-like image properties are probably one of many properties contributing to an aesthetic evaluation. Whether they are necessary or sufficient for an aesthetic evaluation needs to be determined (*Redies, Hasenstein & Denzler, 2007*).

The early differentiation we observed here is in contrast to theories put forward by *Chatterjee (2004)* or *Leder et al. (2004)* who do not assume an influence of aesthetic properties on early processing. More in line with these assumptions are the results by *Höfel & Jacobsen (2007a)* and *Höfel & Jacobsen (2007b)*. They observed a differentiation of ERPs to aesthetic and unaesthetic picture starting at around 300 ms after stimulus onset (see also *Jacobsen & Höfel, 2003* using the same stimuli but a different task for a similar time pattern). The stimuli, *Höfel & Jacobsen (2007a)* and *Höfel & Jacobsen (2007b)* employed, were black-white symmetric and asymmetric patterns instead of colored webpages as we

used. Such stimuli apparently elicit similar ERP-patterns but they might miss properties that the webpages employed here had, for instance, different degree of complexity, color and so on.

Some studies in this area (*Lindgaard et al., 2006*; *Lindgaard et al., 2011*; *Tractinsky et al., 2006*; etc.) have presented stimuli for very brief period of 50 ms duration (*Tuch et al., 2012a* even for 17 ms). Evaluations of these shortly presented stimuli were quite stable—but it is not to be supposed that the cognitive processing of these stimuli only takes 50 ms. Based on our data, we assume that evaluation is a process that lasts for several hundred milliseconds, but can be initiated even with brief presentation durations.

We argued above that experts have enhanced, widely distributed representations that are easy to access. Thus, attention processes operating in a form of a feed-forward sweep might have influenced aesthetic evaluation processes already early on (*Chatterjee, 2004*). However, this would imply an interaction of expertise and webpage aesthetics that was absent here. The aesthetic evaluation is probably based on a variety of stimulus properties that are processed in a bottom-up manner first (*Douneva, Jaron & Thielsch, 2016*; *Thielsch & Hirschfeld, 2012*) before top-down processes kick in. It is rather unlikely, that bottom-up processing affects stimulus processing over the whole analysis period. Rather, evaluative impression formation and evaluative categorization take place in this period (*Cela-Conde et al., 2004*; *Jacobsen & Höfel, 2003*). However, the long lasting difference between aesthetic and unaesthetic webpages prohibits relating cognitive processes to time periods in a fine-grained manner. The spatial distribution of the ERP-effect observed is of little help. Spatial and temporal distributions of ERP-effects allow only relatively gross classification. Evaluative processes might bring about the interaction of aesthetic and anterior–posterior activation in the time-window of 150 ms–600 ms. The occipital activation in the earlier time-window might reflect mainly processing of aesthetic stimulus properties.

Aesthetic webpages elicited more negative ERP than unaesthetic ones. There is a more negative going ERP at central to lateral electrodes in a time-window of 150 ms to 370 ms after stimulus onset for aesthetic compared to unaesthetic webpages. *Cela-Conde et al. (2004)* but also *Jacobsen & Höfel (2003)* observed much more temporally and spatially distinct differences between aesthetic and unaesthetic stimuli than we did. Jacobsen and Höfel observed a more negative going ERP when participants viewed unaesthetic stimuli. However, they compared ''beautiful'' decisions to ''symmetric''-decision. The different temporal and spatial distribution might be due to the employed task. In addition, while we used linked mastoids as reference electrode, Jacobsen and Höfel used the nose tip as reference electrode. That prevents re-referencing our data to their setup. Thus, the differences in spatial distribution might simply be due to reference differences.

*Cela-Conde et al. (2004)* observed effects starting at around 400 ms to 900 ms after stimulus onset while participants observed aesthetic (rated beautiful) stimuli. The late onset might be due to using artistic and non-artistic stimuli instead of ''everyday'' aesthetic and unaesthetic stimuli. Furthermore, the spatial differences probably result from the fact that Cela-Conde et al. used MEG, a reference-free measure and equivalent dipoles in the source space to determine the spatial distribution. The ''relatively'' small number

of electrodes that we used prohibits source-location. In sum, task, stimuli and recording technique might contribute to the observed differences.

## CONCLUSIONS

The relevant aesthetic theories (*Chatterjee, 2004*; *Leder et al., 2004*; *Reber, Schwarz & Winkielman, 2004*) predict an interaction between recipient characteristics and stimulus properties. Leder et al. assume that an aesthetic form should facilitate perceptual and cognitive processing given expertise (see also *Reber, Schwarz & Winkielman, 2004*). More expertise should result in less cognitive effort, hence in less neuronal activation. We could not find such interaction; rather we observed only main effects of recipient characteristics and of stimulus properties. Also not anticipated, experts showed more activation than laypersons. One might assume, that the observed activation reflects the broader and better-connected associative network that experts supposedly develop (*Cheung & Bar, 2012*; *Harel et al., 2010*). Nonetheless, differences between aesthetically and unaesthetically judged webpages emerge much earlier than anticipated.

### Funding
The authors received no funding for this work.

### Competing Interests
The authors declare there are no competing interests.

### Author Contributions
- Jens Bölte conceived and designed the experiments, analyzed the data, contributed reagents/materials/analysis tools, wrote the paper, prepared figures and/or tables, reviewed drafts of the paper.
- Thomas M. Hösker conceived and designed the experiments, performed the experiments, analyzed the data, contributed reagents/materials/analysis tools, wrote the paper, prepared figures and/or tables, reviewed drafts of the paper.
- Gerrit Hirschfeld and Meinald T. Thielsch conceived and designed the experiments, analyzed the data, contributed reagents/materials/analysis tools, wrote the paper, reviewed drafts of the paper.

### Human Ethics
The following information was supplied relating to ethical approvals (i.e., approving body and any reference numbers):

Our study did not require the approval of our local Ethics Committee at the Department of Psychology at the University of Münster as we performed a non-clinical website evaluation study and only non-invasive measures (ratings, reaction times, EEG).

The task was to assess different websites in respect to their aesthetics. No treatments or false feedbacks were given; no potential harmful evaluation methods were used.

Participation was completely voluntary and participants could drop out at any time without any negative consequences. Informed consent was obtained from all participants. All data were stored only using an anonymous ID for each participant.

### Data Availability

The raw data has been supplied as a Supplementary File.

### Supplemental Information

Supplemental information for this article can be found online at http://dx.doi.org/10.7717/peerj.3440#supplemental-information.

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
