# Peer review of "Electrophysiological correlates of aesthetic processing of webpages: a comparison of experts and laypersons"

_PeerJ, doi:10.7717/peerj.3440_

## Round 0.1 · original submission · Major Revisions

Thank you for your patience - it was surprisingly difficult to find appropriate reviewers who were willing to review this paper. However, I think you find that Reviewer 2 has provided a very detailed and thoughtful set of comments, both in their review and in their annotation of your paper. Although there are many comments that will require some work, I believe the feedback can be addressed with appropriate revisions.

Reviewer 1 ·

Basic reporting

"No Comments"

Experimental design

"No Comments"

Validity of the findings

"No Comments"

Additional comments

The manuscript is written in clear and professional English; well structured and organized; however, I have two minor suggestions that I think would make presentation of the results clearer:-

- First, in Table 1 (in the “Results” section, line 275), I think switching the headings of the second raw and the third raw would make the table clearer and less confusing. The second raw should include the two headings (“unaesthetic webpage” and “aesthetic webpage”) and the third raw should include the two pairs of sub-headings of (“aesthetic” and “unaesthetic”).

- Second, in the “EEG-Data" section (starting from line 289), I suggest summarizing the analysis results presented in this section in a table. I think this would help the reader in comparing the results among the various cases and in getting the overall picture.

·

Basic reporting

Reviewer comments
Angela Gosling


Summary- The authors present an interesting paper that seeks to identify both the neural and behavioural correlates of expert aesthetic processing in the context of website evaluation. The authors report differences in the behavioural classification of websites that depended on expert versus lay evaluations (as being aesthetic versus unaesthetic) these differences were driven, in the main, by evaluations for unaesthetic websites. The authors report neural measures of event-related potentials to investigate the time course of expert aesthetic evaluation processes and further whether ERPs provide evidence to support current models of aesthetic evaluation based on interactions between automatic and controlled evaluation mechanisms in the brain (that may differ for experts versus lay individuals).

General comments- I highly recommend this paper for publication following some minor changes (following some minor changes/clarification), the study presents a novel dataset that will be of great interest to researchers working across disciples of Visual Aesthetics, and Neuroaesthetics. The work will also be of interest more generally to those working in the area of visual cognition, cognitive neuroscience and within the field of event-related potentials in particular.

Specific comments
The article was generally well written, although see comments on paper for some minor grammatical errors and sentences that needed some clarification.
The copies of the figures I received do not seem to have the spatial resolution required for publication- but it may be that there are figures that I do not have.
The experimental design is theoretically grounded and sensible given the research questions

Main points that need to be addressed
The predictions/hypothesis need to be more straightforward – the authors provide a rationale for two event related potential components very well (namely EDAN and LPP) they also mention some ERP work investigating effects of expertise- this needs to be built on to provide a clearer rationale for specific components that can be used to test the perceptual and cognitive predictions of the model s of aesthetic processing under scrutiny- please see the large number of comments of have made in the work in this regard
There are also a number of questions that I have raised (as comments in the manuscript to make this easier for the authors to review my comments)
There also seems to be some confusion about the reporting of ERP components – for example the authors discuss the N1 during a specific time window – because analyses is being carried out across the whole head the negativity they refer too is actually a posterior positivity. Please see the notes in the work
It is more usual for ERP analyses carried out during early latencies (sensory components) to focus on particular components that with spatial distributions that allow their functional significance to be discussed- it may be that the authors can used topographic difference maps to show the temporal and spatial distribution of the significant effects they have identified and thereby make their discussion more straightforward in terms of testing hypotheses (in regard to the time course of expert aesthetic processes).

Overall- this is a really interesting paper and I think it is well worth clarifying the points I have highlighted as the results will be of much interest to the research communities working within visual cognition and cognitive neuroscience.

Experimental design

Design is appropriate - see comments in previous section and on manuscript

Validity of the findings

I have made a number of suggestions which may clarify the results/analyses and make interpretation more straightforward (please see the copious notes on the manuscript)

Additional comments

I have made a large number of comments on the manuscript (using comments boxes)
I think this paper only requires some small clarifications before it can go forward for publication- I think the design is really novel and the data are potentially very interesting

---

## Round 0.2 · Minor Revisions

Reviewer 2 has made some clear and relevant comments regarding the description of the experiment and related discussion points. Addressing these comments should be straightforward, and the resulting paper will be much clearer. Please address these points as best possible;

Reviewer 1 ·

Basic reporting

no comment

Experimental design

no comment

Validity of the findings

no comment

·

Basic reporting

This article is focused on a research question that will be of interest to readers across a broad range of disciples- it addresses a highly relevant and very impactful question. I can recommend publication of this paper following some attention to the details outlined below

There are some minor grammatical problems in the manuscript, these are not a large problem but when reading through it was clear the paper would be improved by a very through proof read

Experimental design

the design is appropriate and addresses the research question

Validity of the findings

The query I raised with regard to the number of trials that make up the event related potentials can be seen in the extracts below, it is likely that this is not an error but perhaps simply needs clarification as the two statements appear to be contradictory?

I raised this question in the initial review and I do not think it has been adequately addressed- please see extracts from paper (that seem to contradict each other?)

Extract 1 … The experiment lasted about 25 minutes. In total 150 stimuli (75 aesthetic and 75 unaesthetic), were presented once in random order determined for each participant.
Extract 2 … A trial was considered congruent if there was a congruency of the participant’s aesthetic judgment and the a-priori classification in aesthetic and unaesthetic webpages. Only congruent trials were used to calculate the grand-averages. After removal of trials containing artifacts this resulted in 705 aesthetic (experts: 292, laypersons: 413) and 1136 unaesthetic trials (experts: 541, laypersons: 595) that were averaged.

Additional comments

Reply
The query I raised with regard to the number of trials that make up the event related potentials can be seen in the extracts below, it is likely that this is not an error but perhaps simply needs clarification as the two statements appear to be contradictory? this needs to be made absolutely clear before publication

Extract 1 … The experiment lasted about 25 minutes. In total 150 stimuli (75 aesthetic and 75 unaesthetic), were presented once in random order determined for each participant.
Extract 2 … A trial was considered congruent if there was a congruency of the participant’s aesthetic judgment and the a-priori classification in aesthetic and unaesthetic webpages. Only congruent trials were used to calculate the grand-averages. After removal of trials containing artifacts this resulted in 705 aesthetic (experts: 292, laypersons: 413) and 1136 unaesthetic trials (experts: 541, laypersons: 595) that were averaged.

General comment
The comments made in the initial review with regard to response time (behavioural) data were meant to highlight how RTs, and implicit in this the response preparation and decision making processes associated with longer or shorter latencies, are associated with later ERPs such as P3 and LPP. So even though your design did not target/aim to investigate RTs as such the impact of differences in response time between your groups may have been reflected in the ERP data- I think this is a point for the discussion- for example, if a participant has a delayed response (as in your paradigm) but knows they will be making a behavioural response at the end of the trial, then response preparation will take place of course in the latency prior to response collection and this will be reflected in the neural dynamics and time course of ERP components- this is why i suggested that readers would be interested to know whether there is a statistical difference between the two groups in behavior (it was not necessary to go into too much detail). It is then possible to see how brain and behavior are linked, for example- the main effect of group membership during the time window P3 is likely linked to behvioural differences in response evaluation and resulting in experts had longer RTs (this will influence the P3 in either latency or amplitude), I hope this is clear

There are still some minor grammatical errors, I suggest you give the manuscript to a proof reader for a final and through read

---

## Round 0.3 · accepted · Accept

Thank you for your careful responses to the reviewers' comments.